# Fuel Consumption Reduction and Efficiency Improvement in Urban Street Sweeper Using Power Split with Lockup Clutch Transmission



**Danilo D'Andrea** (ID)**, Giacomo Risitano** (ID) **and Fabio Alberti \***(ID)

Department of Engineering, University of Messina, Contrada di Dio, 98166 Messina, Italy
* Correspondence: fabio.alberti@unime.it

**Abstract:** The aim of this work is to design a power split transmission for an urban street sweeper in order to reduce fuel consumption. The design process starts with the comparison between a hydrostatic and a hydromechanical power split transmission. Both transmissions have been tested through an acceleration test considering 30, 50, 70 and 100 percent of the rated engine power. The results of both models developed in the Simcenter Amesim$^{TM}$ environment show that the power split transmission presents a higher efficiency, which justifies the adoption of this type of transmission with respect to the hydrostatic system. Then, a pure mechanical gear is added to the base concept of the power split transmission. The mechanical gear is managed by a lockup clutch, which can be engaged during the working phase of the street sweeper, similar to an adaptive cruise control. In this case, both transmissions are tested through a regulated cycle, UNI-EN 151429-2, highlighting the advantage of using a pure mechanical branch. At the end, both transmissions are tested with a driving cycle acquired through an experimental setup consisting of a control unit, a GPS and a tablet for the monitoring of the speed profile. The results show that the adoption of a lockup clutch allows an increase in the system efficiency during the working phase, hence reducing the average fuel consumption during the mission test.

**Keywords:** street sweeper; hydrostatic transmission; hydromechanical transmission; efficiency improvement; fuel consumption reduction

## 1. Introduction

The issue of noise and environmental pollution currently represents one of the main themes for the improvement of the quality of human health in cities [1–3]. Several studies have been conducted focusing on the development of green and sustainable mobility [4,5], and on the reduction of the fuel consumption of urban vehicles such as buses or cars [6–8], or of vehicles with high power density, such as wheel loaders or agricultural tractors [9–11]. Moreover, remarkable attention concerns the reduction of emissions from vehicles operating in a city environment, such as taxis [12,13] and other urban vehicles [14]. Despite full electric technology becoming increasingly popular in the automotive field, working machine technology, such as tractors and excavators, is far from the large-scale adoption of full electric architecture [15,16]. In this way, the adoption of a hydromechanical transmission for this type of industrial application could represent a valid alternative for increasing the system efficiency and reducing fuel consumption and emissions, waiting for the reliable and less expensive solutions of full electric technology. In vehicles that use hydraulic/hydromechanical transmissions, the efficiency of the system is linked to the hydraulic pump/motor and to the issues related to the friction and wear that cause several failure modes such as delamination, seizure, noise, vibration and fatigue cracks [17]. As reported in different scientific studies [18,19], one of the solutions may be the employment of anti-friction and nanostructured coatings aimed at improving efficiency and reducing the failure risk. In addition, as reported in recent studies [20], crack propagation and fatigue failure are related to the type of propulsion, e.g., hydraulic and hydromechanical,

such as those investigated is this work. The aim of this paper is to assess the energy efficiency of a mini street sweeper in order to reduce fuel consumption and emissions. Other studies have already dealt with the issue of fuel consumption and emissions of a street sweeper adopting a full electric transmission configuration [21,22]. Despite the advantages related to the reduction of fuel consumption and emissions, working machine technology is currently far from adopting a full electric architecture that also employs artificial intelligence and neural networks [23]. This fact is related to a higher complexity system (electric motor/generator, power converters, control systems and so on), which leads to lower reliability and increased costs. Hence, a hydromechanical transmission could represent a valid alternative for reducing fuel consumption and emissions while waiting for reliable and less expensive fully electric technology. Hydrostatic transmission systems have been used for several years in all hydraulic applications that require fluid energy to perform their functions. They are used in various industrial and automotive applications such as road vehicles, earth-moving machines, agricultural machines and in large part in industrial machines, especially in vehicles operating in severe working conditions [24–26]. In the last decade, many studies have been carried out on the efficiency of hydraulic transmissions due to the increasing demand to reduce fuel consumption and for comfort efficiency, as well as the need to increase the maximum speed in the heavy off-road vehicle sector, where faster road driving speed is desired [27,28]. In the overview of hydraulic power transmissions, attention is focused on the CVT (continuously variable transmission) power split that combines the variability of the CVT with the efficiency of a mechanical transmission [29]. A high-efficiency mechanical branch and a hydrostatic one, which allows the continuous variation of the gear ratio between two fixed points, characterize it. This kind of transmission enables dividing the motor rotation speed with respect to the shaft rotation speed at the transmission output. CVT power splits are structured according to three-shaft configurations, known also as input-coupled (IC) and output-coupled (OC), or to four-shaft configurations, known as dual-stage and compound [30,31]. Although a pure mechanical transmission with gearbox has higher efficiency than a power split transmission, it would make the use of the streetsweeper uncomfortable. For this reason, the transmission adopted for the streetsweeper under consideration follows the CVT power split input couple (IC) configuration. Considering that the use of the vehicle requires long periods of operation at almost constant speed, in order to improve the efficiency of the transmission, a mechanical gear engaged by lockup clutch was designed to bypass the CVT branch. The literature [32,33] shows that the lockup clutch leads to an improvement of the efficiency and fuel economy of the system. The novelty of this study concerns the design of a CVT power split hydromechanical transmission for small machines, such as a mini street sweeper. In this study, a comparison was first made between a hydrostatic transmission commonly used on these vehicles, with a power split input-coupled transmission typically used on vehicles with high power density. The advantage of using the IC has been demonstrated from the point of view of emissions compared to the hydrostatic transmission. However, the street sweeper in the working phase moves in a speed range that does not allow the maintenance of the full mechanical point (FMP) and therefore of maximum efficiency. To overcome this problem, the authors designed a power split input-coupled transmission with the addition of a lockup clutch. The lockup clutch control system is designed to use a mechanical gear during the sweeper's operation and the CVT power split transmission during urban racing, improving overall efficiency and fuel consumption. The design of the transmission and the evaluation of the efficiency and fuel consumption were carried out using Simcenter Amesim$^{TM}$ simulation software. The first stage of the work was the design and comparison of the hydraulic transmission and the CVT split transmission. The simulation result shows greater efficiency for the power split CVT transmission, which justifies the use of the transmission for this application, despite the increased complexity of the system. Finally, a comparison was made between a power split CVT and a power split CVT with lockup clutch to highlight the advantage of using a mechanical gearbox during the working phase of the vehicle, in terms of efficiency and fuel consumption.

## 2. Materials and Methods

### 2.1. Reference Vehicle

The reference vehicle is a Dulevo sweeper 850 mini, whose overall dimensions are reported in Figure 1.

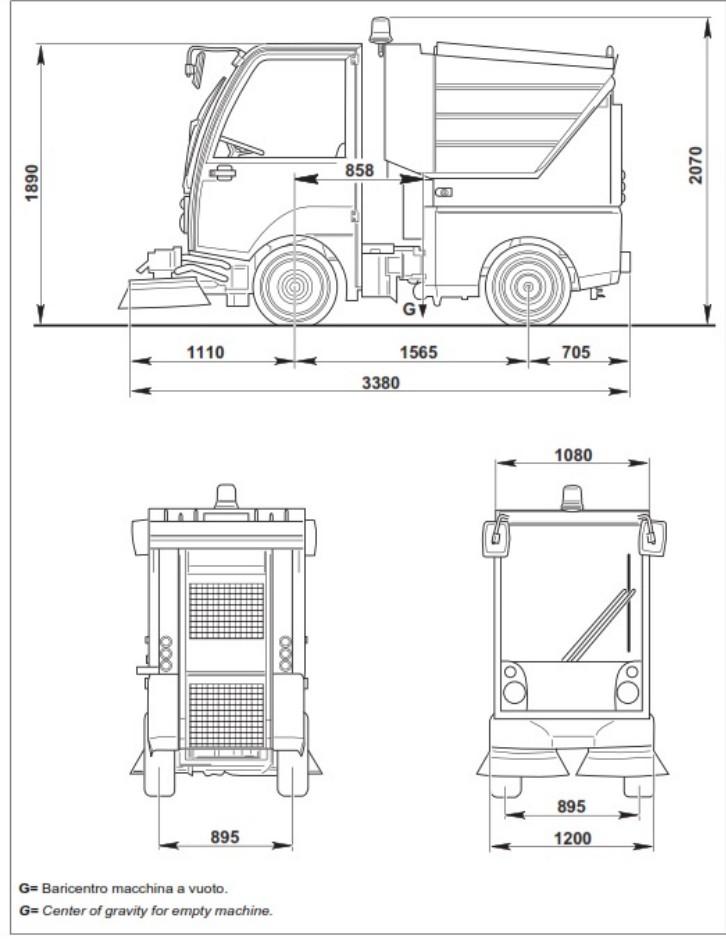

**Figure 1.** Overall dimensions of Dulevo sweeper 850 mini.

The main features of the street sweeper are shown in Table 1.

**Table 1.** Operating characteristics of reference vehicle.

| Parameter | Value |
| --- | --- |
| Engine power ($P_{ICE}$) | 25 kW |
| Engine speed ($\omega_{ICE}$) | 1800 rpm |
| Maximum wheel torque ($C_{wheel}$) | 4200 Nm |
| Maximum vehicle speed ($V_{max}$) | 25 km/h |
| Vehicle mass (m) | 2400 kg |
| Wheel radius ($r_{wheel}$) | 0.35 m |

The internal combustion engine (ICE) adopted is a diesel inline four-cylinder natural aspirated capable of developing a maximum torque of 92.6 Nm at 1700 rpm. The break-specific consumption map is shown in Figure 2.

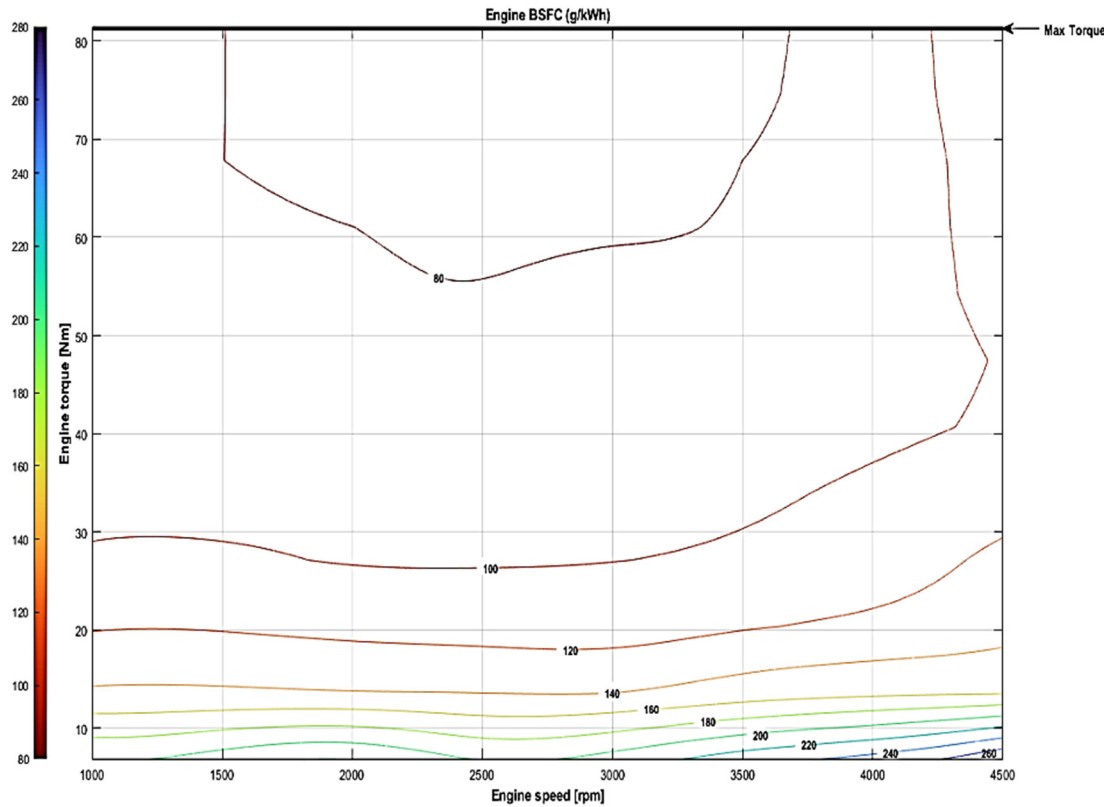

**Figure 2.** BSFC map of ICE engine.

*2.2. Powertrain Layout*

The overall efficiency and fuel consumption of the street sweeper was assessed by comparing the application of three different transmission layouts:

1. Hydromechanical (HM) power split input coupled transmission.
2. Hydrostatic transmission.
3. Hydromechanical (HM) power split input-coupled transmission with lockup clutch.

The design process used to size the three drivelines was as follows. The calculations proposed by Carl et al. [34] were considered for the sizing, which makes a comparison between the four main architectures of the CVT power split. In addition to the reference vehicle parameters, it is necessary to consider some parameters that represent constraints for the design of the transmission, as shown in Table 2.

**Table 2.** Hydrostatic and mechanical system parameters.

| Parameter | Value |
|---|---|
| Max pressure difference ($\Delta_p$) | 380 bar |
| Maximum unit speed ($\omega_{motor/pump}$) | 3500 rpm |
| Full mechanical point (FMP) | 10 km/h |
| Differential transmission ratio ($\tau_{diff}$) | 4 |
| Planetary transmission ratio ($\tau_0$) | $-1/3$ |

The preliminary design of a hydromechanical transmission derives mainly from the choice of the standing gear ratio $\tau_0$ of the planetary gear, the differential gear ratio $\tau_{diff}$ and the speed at which the transmission will reach the full mechanical point, as shown in Table 2. In order to design the input-coupled transmission (Figure 3), the first parameter to choose is the vehicle speed at which the full mechanical point (FMP) occurs. The FMP is theoretically the point where the maximum efficiency of the transmission occurs. As

suggested by Rossetti and MacOr [35], the FMP is set to 1/3 of the maximum vehicle speed—in this case, 10 km/h.

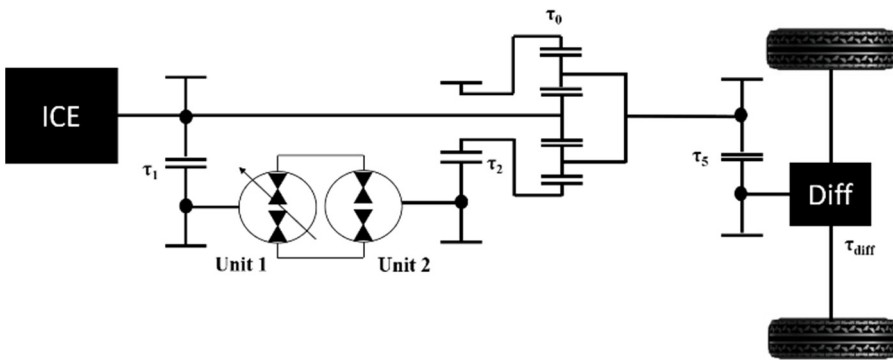

**Figure 3.** Scheme of input-coupled transmission.

The FMP occurs when the speed of the planetary crown C is zero, i.e., when the speed of the shaft of unit II (hydraulic motor) is zero. The vehicle speed in the FMP depends on the choice of the transmission ratio to the differential $\tau_{diff}$, the transmission ratio of the planetary gear when the crown C has zero speed $\tau_0$. When the transmission ratio of the planetary gear $\tau_0$ and the transmission ratio of the differential gear $\tau_{diff}$ are arbitrarily chosen and the speed at which the FMP will occur is defined, through (1), it is possible to determine the transmission ratio $\tau_5$, which represents the transmission ratio assumed by the gearing downstream of the planetary gear, and upstream of the differential box:

$$V_{FMP} = \frac{\omega_{ICE}}{(1 - \tau_0) \cdot \tau_{diff} \cdot \tau_5} \cdot r_{wheel} \cdot 3.6 \cdot \frac{\pi}{30} \tag{1}$$

The transmission ratio $\tau_2$ between the shaft of the second unit (hydraulic motor) and the crown of the planetary gear is defined by setting the maximum speed of unit II as the boundary condition, according to (2):

$$\tau_2 = \omega_{motor} \cdot \tau_0 \cdot \left[ \frac{V_{max} \cdot \tau_{diff} \cdot \tau_5}{r_{wheel} \cdot 3.6} \cdot (\tau_0 - 1) \cdot \frac{30}{\pi} + \omega_{max} \right]^{-1} \tag{2}$$

The displacement of the second hydraulic unit is determined by imposing the maximum pressure limit $\Delta p$, and the maximum wheel torque $C_{wheel}$, according to (3):

$$V_{II} = C_{wheel,max} \cdot \left( 1 - \frac{1}{\tau_0} \right)^{-1} \cdot (\Delta_p \cdot \tau_2 \cdot \tau_{diff} \cdot \tau_5)^{-1} \cdot 20\pi \tag{3}$$

The transmission ratio $\tau_1$ is determined by the ratio between the speed of the internal combustion engine shaft ($rpm_{ICE}$) and the rotation speed of the pump shaft, imposing the maximum pump speed as the boundary condition, according to (4):

$$\tau_1 = \frac{\omega_{ICE}}{\omega_{pump}} \tag{4}$$

Finally, it is possible to determine the displacement of the first hydraulic unit through (5):

$$V_I = V_{II} \cdot \tau_1 \cdot \tau_2 \cdot (\tau_0)^{-1} \tag{5}$$

Regarding the hydrostatic transmission and the HM with lockup clutch, Figures 4 and 5 shows the transmission schemes.

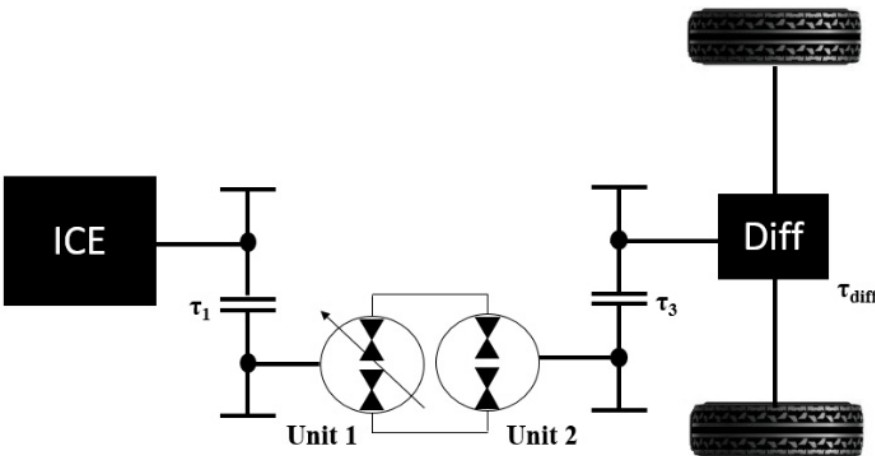

**Figure 4.** Scheme of hydrostatic transmission.

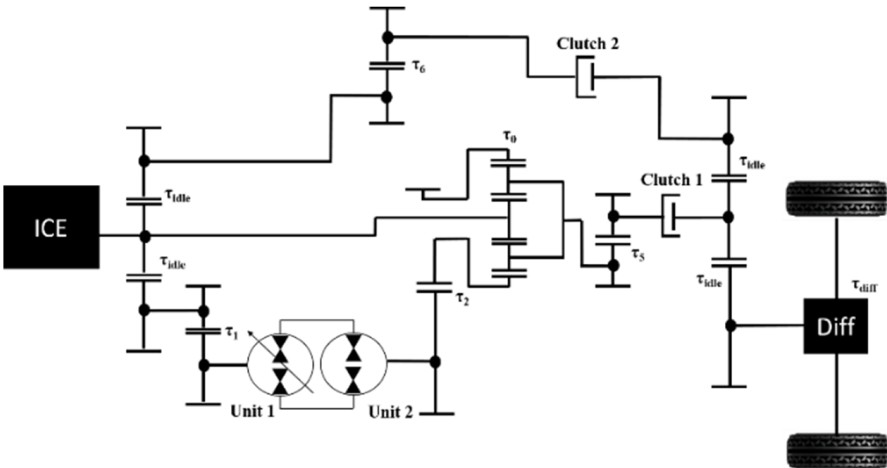

**Figure 5.** Scheme of CVT input-coupled transmission with lockup clutch.

The results of the design calculations are summarized in Table 3.

**Table 3.** Design calculation parameters.

| | | Hydrostatic | CVT IC | CVT IC with Lockup Clutch |
|---|---|---|---|---|
| Parameter | | *Value* | *Value* | *Value* |
| Gear ratio | $\tau_1$ | *0.51* | *0.51* | *0.51* |
| Gear ratio | $\tau_2$ | | *0.32* | *0.32* |
| Gear ratio | $\tau_3$ | *2.41* | | |
| Displacement | $V_I$ | *40 cc* | *40 cc* | *40 cc* |
| Displacement | $V_{II}$ | *39 cc* | *39 cc* | *39 cc* |
| Gear ratio | $\tau_5$ | | *4.52* | *3.77* |
| Gear ratio | $\tau_{diff}$ | *4* | *4* | *4* |
| Gear ratio | $\tau_0$ | | *1/3* | *1/3* |
| Gear ratio | $\tau_6$ | | | *6* |
| Gear ratio | $\tau_{idle}$ | | | *1* |

### 2.3. Numerical Models and Control Strategies

Each component of the transmission, both hydraulic and mechanical, as well as the control system were modeled using Simcenter Amesim$^{TM}$, which allows the creation of a functional transmission scheme using logical connections of mathematical blocks. Through the time variant analysis, it calculates the fundamental parameters of the transmission,

based on mathematical models proposed in different scientific studies [35,36]. To obtain reliable results from the analyses obtained in this study, it was necessary to accurately model the power losses in the transmission, and in particular the volumetric and hydromechanical losses in hydraulic machines. Considering the operation of the input-coupled transmission, a pump, connected to the ICE engine (prime mover), generates flow to drive a hydraulic motor, which is connected, through mechanical gearing, to the load (vehicle). Hence, it was necessary to correctly model the control strategy of the pump displacement, which allows the vehicle to follow the mission test. In order to characterize both the hydrostatic and hydromechanical transmission, different simulations with load partialization were performed, controlling a mechanical friction placed at the end of the transmission. Finally, a control strategy of the lockup clutch was modeled in order to engage only the mechanical branch during the working phase of the street sweeper.

### 2.3.1. Volumetric and Hydromechanical Losses

The volumetric and hydromechanical losses were modeled using the nominal efficiencies and flow rate of a commercial hydraulic pump/motor of 45 cc displacement. This choice was made, with respect to the design condition, in order to guarantee the power transmission in the hydraulic circuit, considering the losses of the real system. Hence, from the flow rate and volumetric efficiency, it is possible to evaluate the leakage losses, as follows:

$$L_{leakage} = L_{nom}\left(1 - \eta_{vol}\right) \tag{6}$$

In the model, leakage losses are taken into account through a hydraulic node in the system, which subtracts the leakage loss from the theoretical flow rate. Figures 6 and 7 show the volumetric efficiency and the nominal flow rate of the hydraulic pump/motor, respectively.

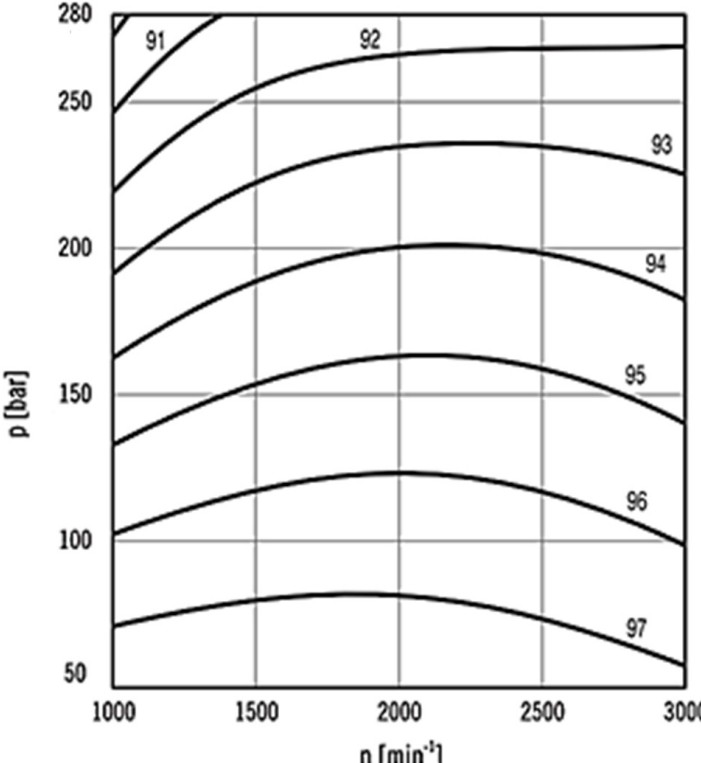

**Figure 6.** Volumetric efficiency of hydraulic pump/motor function of pressure and rotational speed condition.

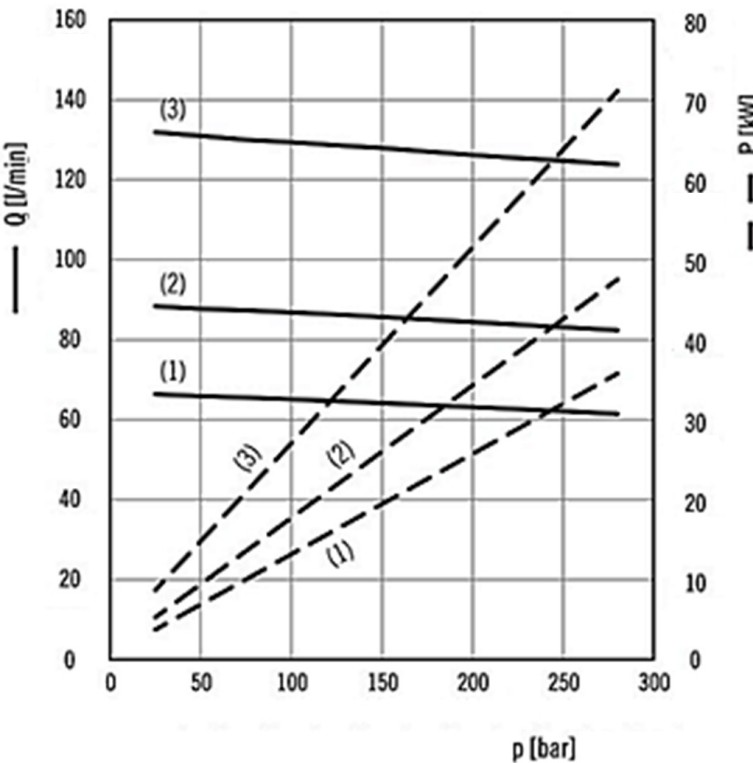

**Figure 7.** Nominal flow rate function of pressure, power and rotational speed. Curves are parameterized as (1) 1500 rpm, (2) 2000 rpm, and (3) 3000 rpm.

Similarly, the mechanical losses were evaluated as follows:

$$L_{mech} = T_{nom} \cdot \eta_{hydro-mech} \tag{7}$$

The mechanical losses are taken into account through a mechanical friction, which subtracts the mechanical loss from the theoretical torque $T_{nom}$. Moreover, the hydromechanical efficiency was calculated as follows:

$$\eta_{hyro-mech} = \frac{\eta_{tot}}{\eta_{vol}} \tag{8}$$

where $\eta_{tot}$ is the total efficiency of hydraulic pump/motors (Figure 8).

### 2.3.2. Pump Displacement Control Strategy

Changing the position of the swash plate will represent changes in the flow direction and flow volume. The direction and displacement of a closed-loop piston pump is handled by a servomechanism, which is basically any self-correcting control system. The system needs to correct any time the operator requests an output from the pump that differs from what is currently being delivered. The operator requests, in terms of acceleration or deceleration, are satisfied by controlling the swash plate angle of the axial piston pump. In order to follow the mission test, the difference between the target velocity and the actual linear velocity of the vehicle is sent to a PID controller. The output of the controller is sent to the pump characteristic that, in the function of the solenoid current, returns the fractional displacement necessary to follow the speed profile. Hence, the control of the fractional displacement allows the control of the vehicle's linear velocity. Figure 9 shows the control strategy adopted to control the pump displacement.

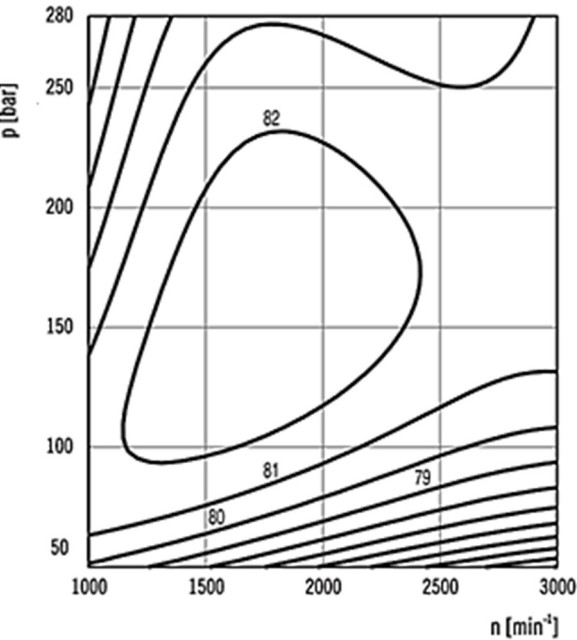

**Figure 8.** Total efficiency of hydraulic pump/motor function of pressure and rotational speed condition.

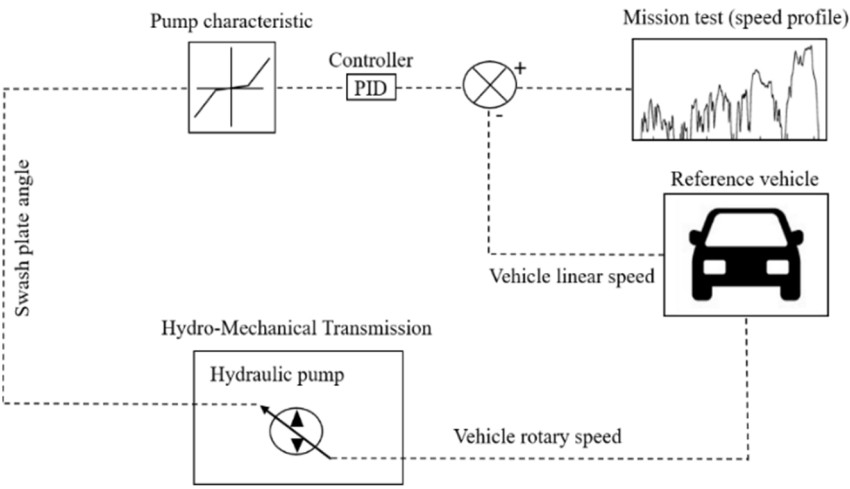

**Figure 9.** Pump displacement control strategy.

### 2.3.3. Lockup Clutch Dynamic Control

The lockup clutch allows the hydromechanical transmission to engage in a pure mechanical branch during the working mode of the streetsweeper, i.e., the low speed, fan, brush motors and water pump are activated. The activation is carried out manually by the operator controlling a mechanical clutch that is able to engage or disengage a mechanical branch. Once the lockup speed has been set, if the vehicle speed value is close to the lockup speed, the mechanical branch remains engaged. On the other hand, if the vehicle accelerates or brakes roughly with respect to the lockup speed, the control system disengages the mechanical branch and the transmission returns to the hydromechanical mode. Hence, the control system works in a similar way to an adaptive cruise control. Considering the greater efficiency of the mechanical branch with respect to the hydromechanical one, the control system improves the global efficiency of the system. Figure 10 shows the lockup clutch control system.

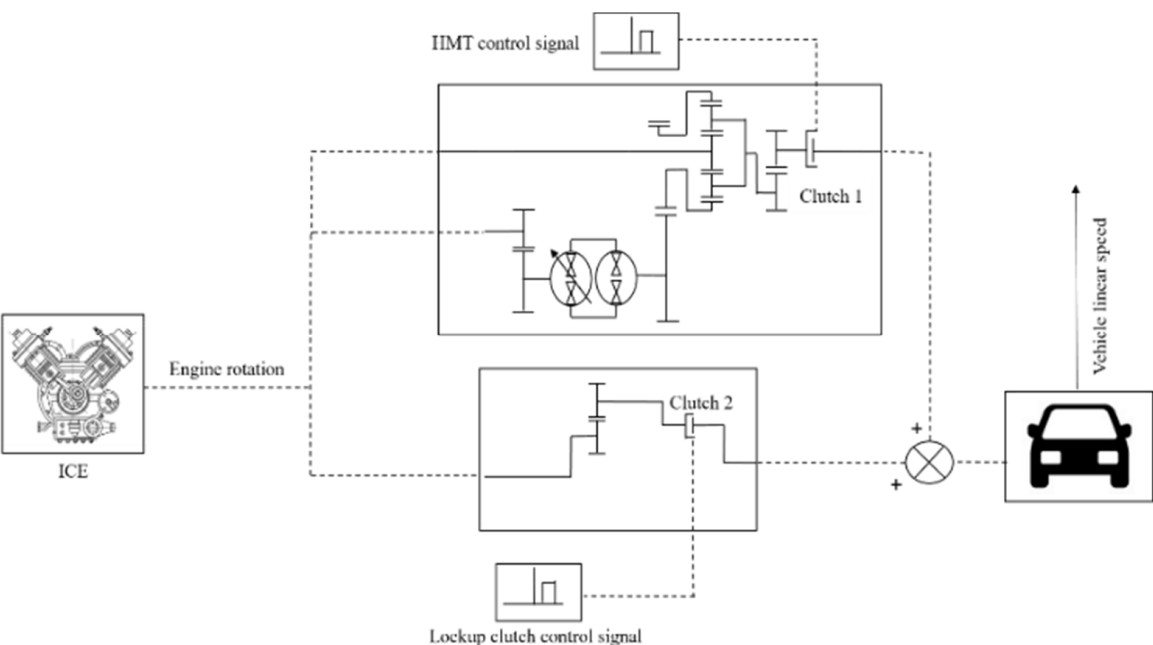

**Figure 10.** Operation mode control signals.

Table 4 shows the control signals implemented in the model.

**Table 4.** Control signal of the operating mode.

| Mode | Value | Power Transmission |
|------|-------|--------------------|
| Urban driving | 1 | Hydromechanical |
| Working | 0 | Mechanical |

In Table 4, a value of 1 means clutch 1 is closed and clutch 2 is open (hydromechanical transmission), while signal 0 means clutch 1 is open and clutch 2 is closed (mechanical transmission).

### 2.3.4. Transmission Characterization

In order to characterize both the hydraulic and hydromechanical transmission, four different simulations using 100%, 70%, 50% and 30% of the rated engine power were performed.

The ICE engine power is controlled imposing a traction force characteristic for the different cases. The maximum traction force is given by (9):

$$F_{tmax} = m \cdot g \cdot \mu = 11760 \text{ N} \tag{9}$$

where $\mu$ is a utilization coefficient set to 0.5. Considering that $F = P/v$, for a lower speed value, the traction force is equal to the maximum traction force, while for a higher speed value the traction force decreases following a hyperbolic trend. Figure 11 shows the imposed traction force in the function of velocity.

The power partialization is imposed onto the system controlling a rotary friction placed at the of the system, after the differential gear. Basically, the rotary friction acts like a variable load depending on the percentage of nominal power (or traction force) imposed to the system. Also in this case, similar to the pump displacement control, a PI controller has been implemented into the model in order to allow the system to follow the imposed traction force. Figure 12 shows the control system used for the transmission characterization.

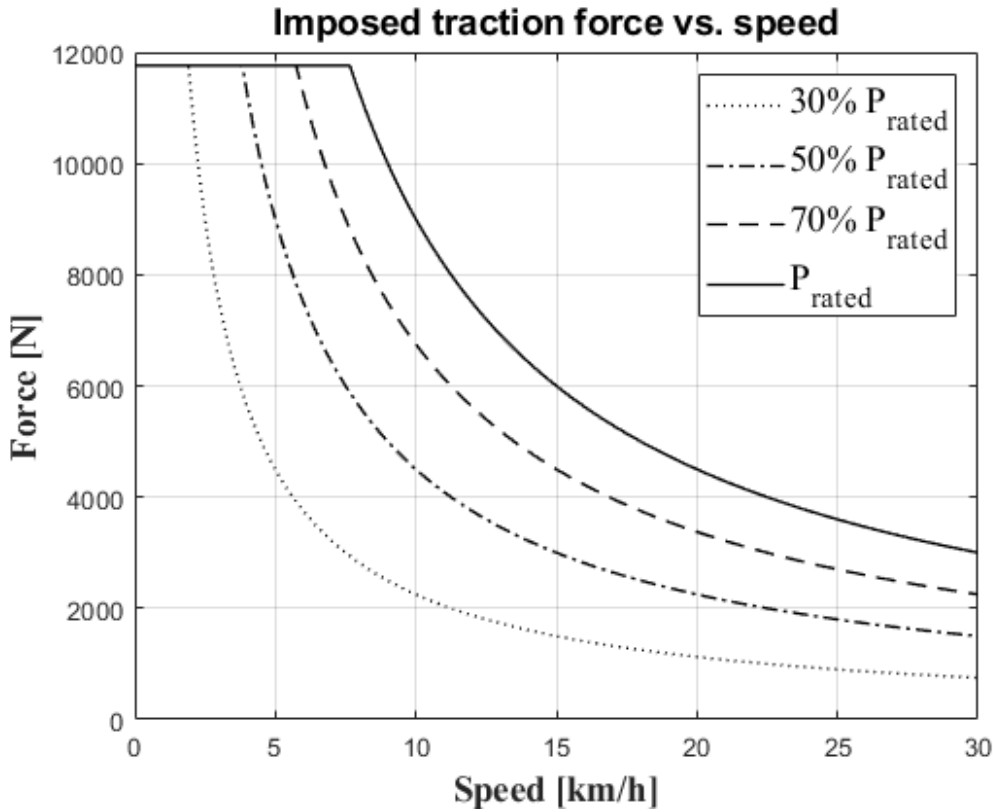

**Figure 11.** Imposed traction force for transmission characterization.

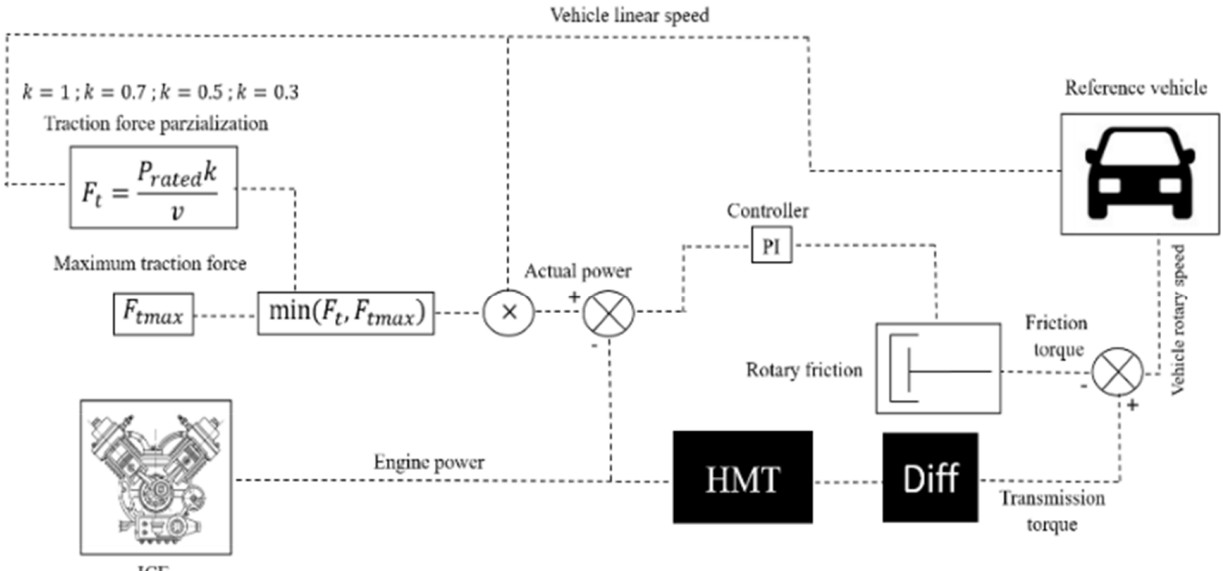

**Figure 12.** Power partialitation control system.

### 2.3.5. UNI-EN 15429 Cycle

Figure 13 shows the standard UNI EN 15429-2 working cycle. It is applied to surface-cleaning machines, with applications in open public areas, roads, airports and industrial complexes.

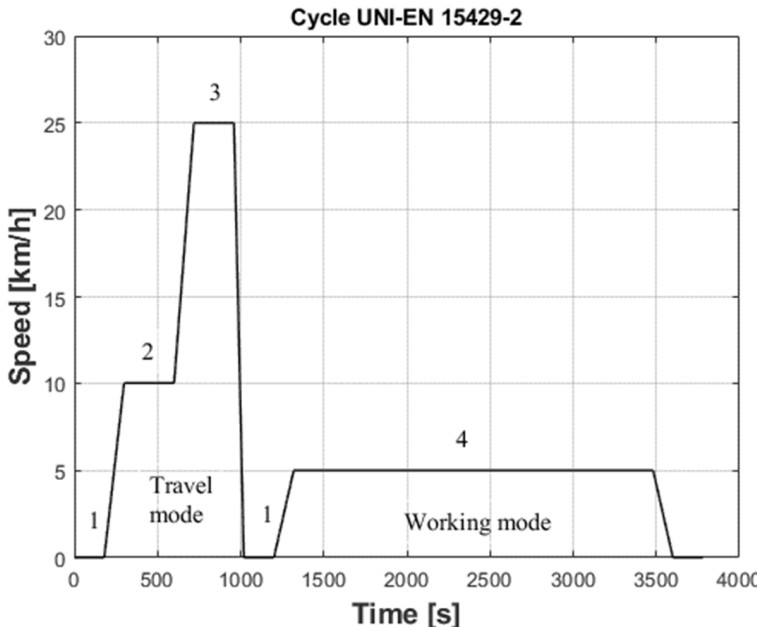

**Figure 13.** Cycle UNI-EN 15429. 1: idle speed, 2: FMP speed, 3: maximum speed, 4: working-mode speed.

### 2.3.6. Mission Test

The transmissions were tested on a real driving cycle acquired in the city of Messina. The instrumentation used consists of a control unit, a SCADAS XS compact model, provided by Siemens^TM, together with a GPS and a tablet for monitoring the channel acquisition. Figure 14a,b show the instrumentation used and the acquired driving cycle, respectively.

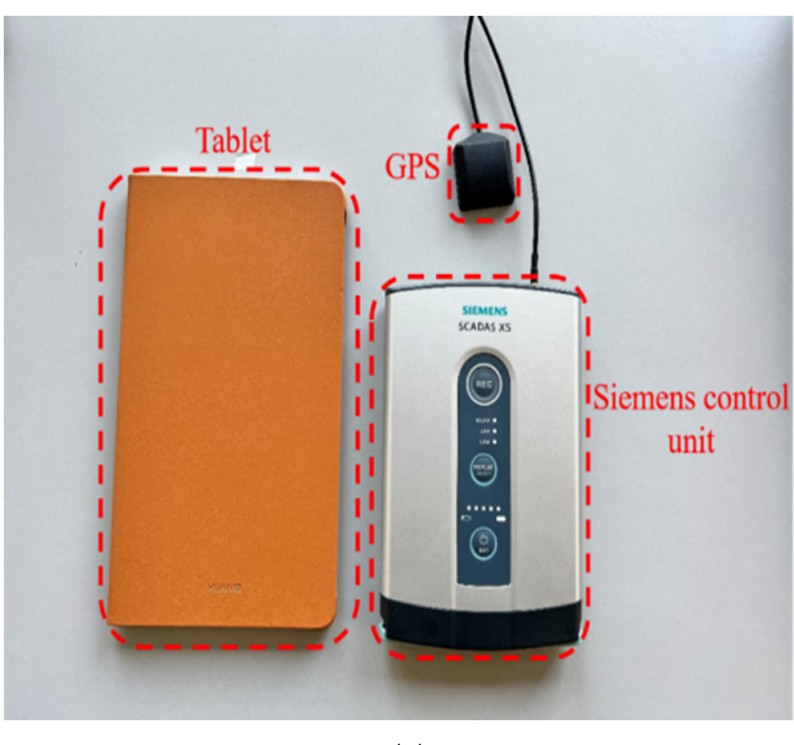

**(a)**

**Figure 14.** *Cont.*

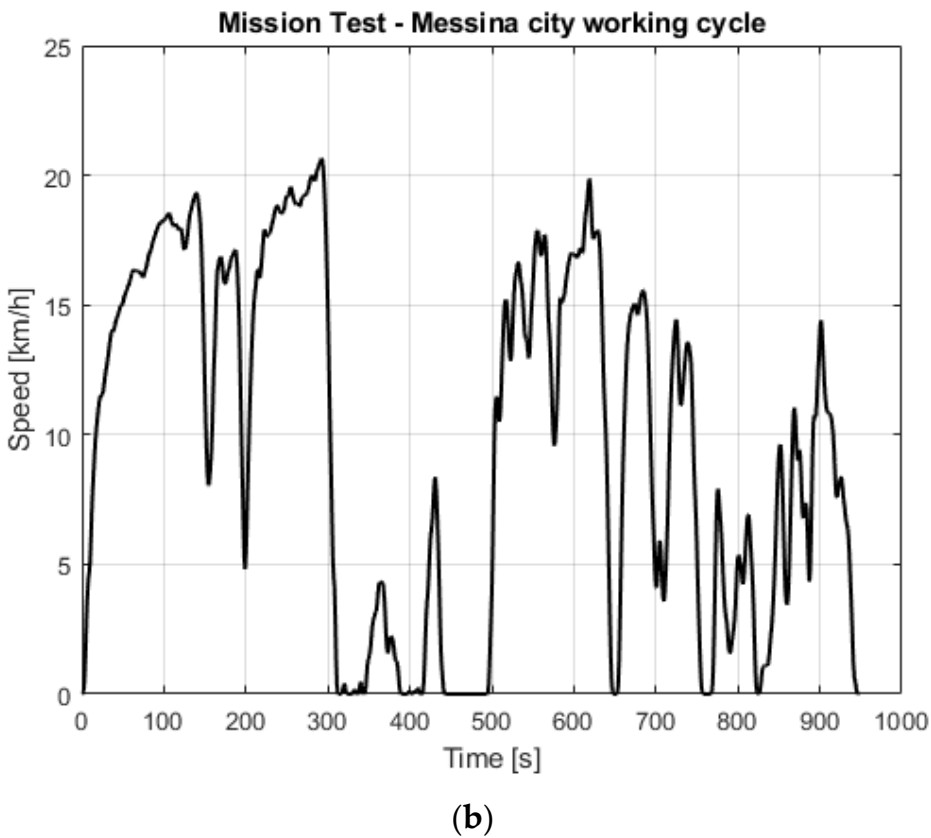

**(b)**

**Figure 14.** Mission test. (**a**) Instrumentation used. It consists of a control unit, GPS and a tablet for monitoring. (**b**) Speed profile.

## 3. Results and Discussion

The first analysis concerns the characterization of both hydrostatic and hydromechanical transmissions with different load levels, as mentioned in Section 2.3.4. Through the characterization, it is possible to compare the transmissions' behavior in terms of overall efficiency. Finally, in order to highlight the advantage of the lockup clutch related to the efficiency and fuel consumption, a comparison between the input-coupled HMT and input-coupled HMT with lockup clutch is carried out by testing the transmissions both on a standard cycle UNI-EN 15429-2 and on a real driving cycle.

### 3.1. Hydraulic Transmission and HMT Input-Coupled Transmission Characterization

In order to characterize both the hydraulic transmission and input-coupled HMT, different simulations were performed with different load levels comparing the global efficiencies. Figures 15 and 16 show the global efficiency $\eta = \frac{P_{out}}{P_{in}}$ for the hydraulic transmission and input-coupled HMT, respectively, in the function of vehicle linear velocity with different load levels. The test consists of an acceleration ramp from zero to the maximum vehicle speed.

The comparison between the hydraulic and input-coupled HMT global efficiency shows a better efficiency of the input-coupled HMT with respect to the hydraulic transmission. This result is related to the hydraulic losses, which are lower in the input-coupled HMT because of the presence of the mechanical branch for the power split transmission. In fact, as shown in Figure 16, the higher efficiency value is achieved at the full mechanical point (FMP), at which the transmission work is purely mechanical. However, in the hydraulic transmission, the power transmission is only related to the hydraulic circuit and hence the hydraulic losses are higher.

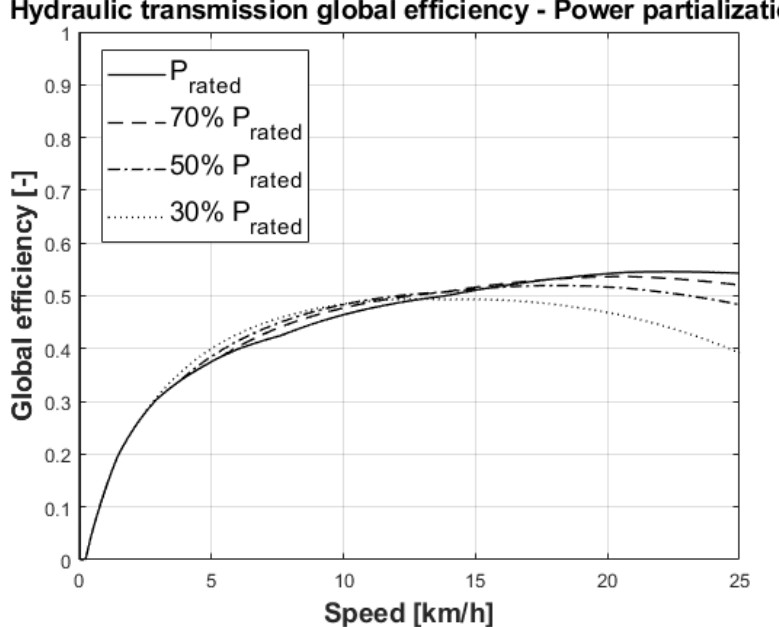

**Figure 15.** Hydraulic transmission global efficiency.

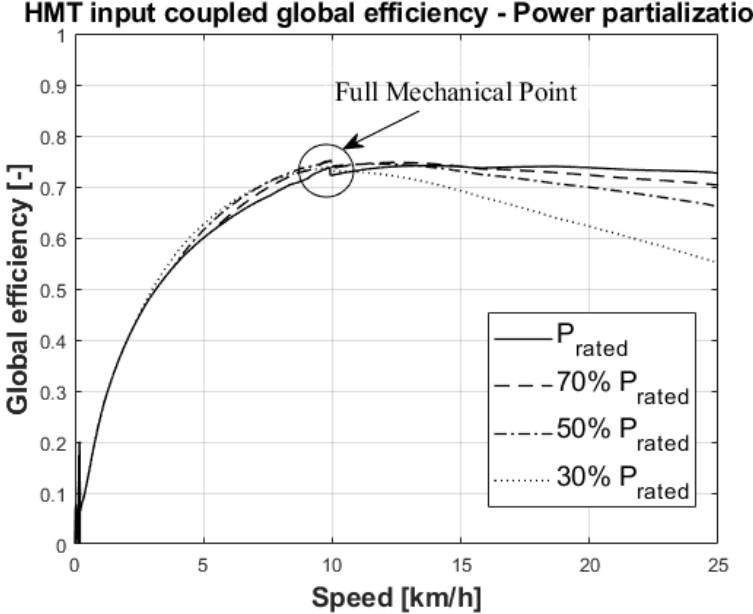

**Figure 16.** HMT global efficiency. Highlighted is the FMP at which the transmission reaches the best efficiency.

Moreover, the results show, for both transmissions, that the power partialization leads to a decrease in the global efficiency with the increase in speed.

Hence, if the application justifies the increased costs of the input-coupled HMT related to a more complex system in terms of planetary gear and control system, the input-coupled HMT turns out to be the most advantageous.

*3.2. Comparison between Input-Coupled HMT and Input-Coupled HMT with Lockup Clutch—UNI EN 15429-2*

In order to highlight the advantage of using a lockup clutch in the input-coupled HMT, the global efficiency and fuel consumption were analyzed for both architectures. At first, the comparison was made through a regulated cycle of UNI EN 15429-2. In this

case, the control strategy allows the mechanical branch to be engaged during the working mode. The working speed settled at 5 km/h, according to Figure 13. Figure 17a,b show the comparison between the global efficiency and fuel consumption of the input-coupled HMT and the input-coupled HMT with lockup clutch, respectively.

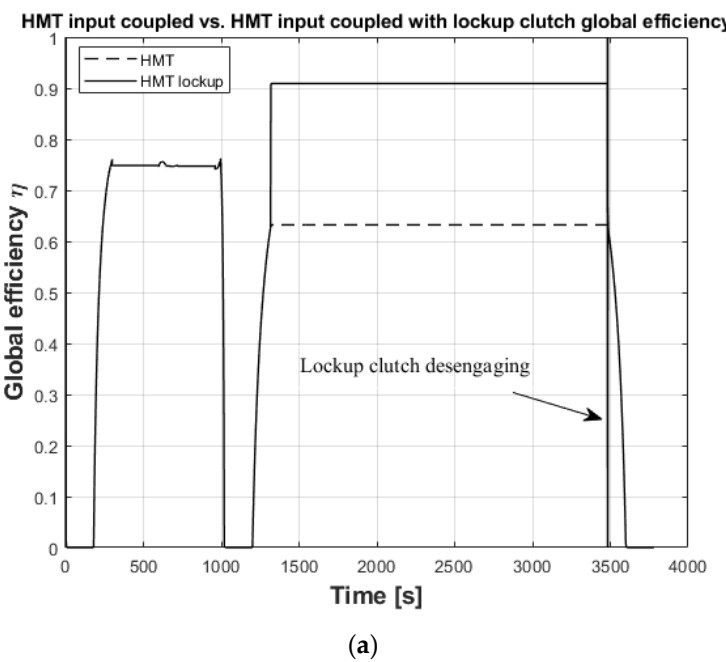

(**a**)

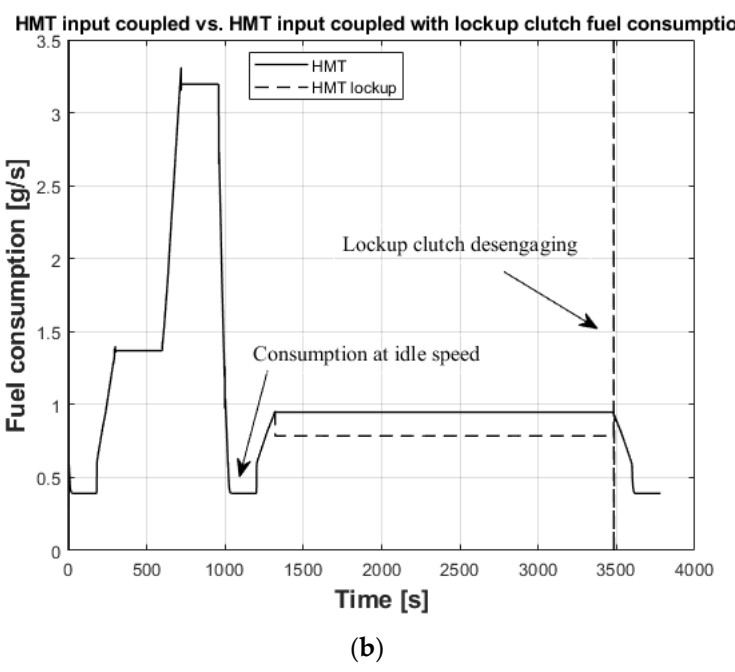

(**b**)

**Figure 17.** Comparison between input-coupled HMT and input-coupled HMT with lockup clutch through UNI-EN 15429-2 cycle. (**a**) Global efficiency. (**b**) Fuel consumption.

The comparison of the global efficiencies shows that the use of a mechanical branch during the working mode leads to a consistent improvement. In fact, as shown in Figure 17a, during the working mode, the global efficiency for the HMT with lockup clutch reaches the value of 0.9, while this value is 0.6 for the power split. The efficiency improvement is also reflected in the fuel consumption, as shown in Figure 17b. The results are summarized in Table 5.

**Table 5.** Comparison between power split and power split with lockup clutch through the standard UNI-EN 15429-2 cycle.

|  | Power Split | Power Split Lockup |
|---|---|---|
| Global efficiency | 0.63 | 0.91 |
| Fuel consumption | 0.95 g/s | 0.7 g/s |

*3.3. Comparison between Input-Coupled HMT and Input-Coupled HMT with Lockup Clutch—Mission Test*

Both transmissions were also tested on a real driving cycle. Figure 18a,b show the comparison between the global efficiency and the fuel consumption, respectively.

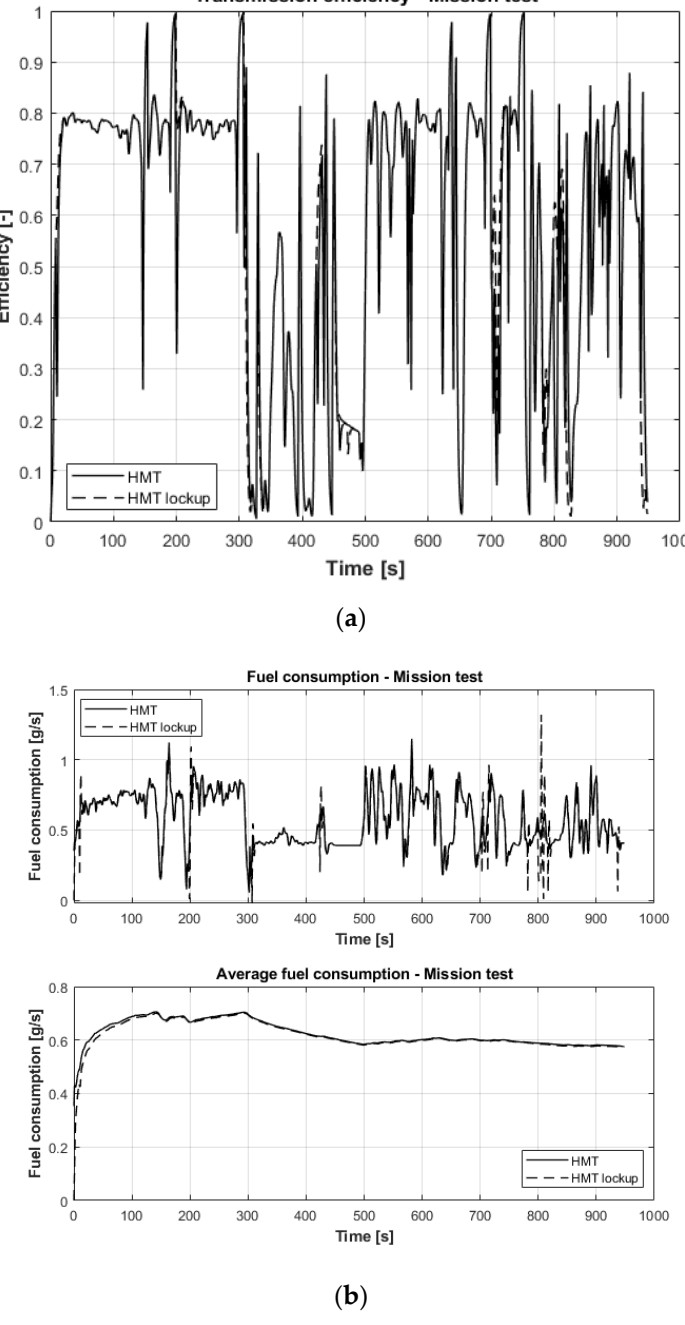

(**a**)

(**b**)

**Figure 18.** Comparison between input-coupled HMT and input-coupled HMT with lockup clutch on a real driving cycle. (**a**) Global efficiency. (**b**) Fuel consumption.

The comparison between the global efficiencies of the input-coupled HMT and the input-coupled HMT with lockup clutch proves that the use of the lockup clutch leads to an increase of the global efficiency of the system when the lockup clutch is engaged (Figure 17a). Moreover, the improvement of the efficiency leads to a decrease in the fuel consumption, as shown in Figure 17b. In this case, the control system allows a mechanical gear to be engaged in the range speed between 4 km/h and 6 km/h, in a similar way to an adaptive cruise control. Hence, in this phase, the vehicle follows the mission test, changing the angular speed of the engine and not controlling the pump displacement, like in power split mode. It is necessary to observe that, with respect to the standard cycle and taking into account a real driving cycle (mission test), the fuel consumption reduction becomes less evident due to the dynamics of a real cycle. However, if the projected daily working cycle is taken into consideration, for example, in one year of work, and also considering the high number of streetsweepers operating in an urban center, the reductions in the fuel consumption and emissions could become considerable. In addition, this advantage in terms of efficiency and fuel consumption in achieved through a simple solution from a design point of view, only involving the addition of a mechanical gear. Hence, this fact could justify the adoption of the lockup clutch.

## 4. Conclusions

In this work, a power split transmission was designed for a city street sweeper in order to improve the global efficiency of the system and reduce the fuel consumption. First, a comparison was made between the hydraulic and input-coupled HM transmission through a power partialization of the ICE engine in order to characterize both transmissions. The results show the higher efficiency of the HMT, which justifies the greater complexity of the system for this type of application. Finally, a comparison between the input-coupled HMT and input-coupled HMT with lockup clutch was performed, testing the street sweeper both on a standard cycle and on a real driving cycle. The lockup clutch allows, through an appropriate control system similar to an adaptive cruise control, a pure mechanical gear to be engaged during the working phase. The results show that through this solution, is possible to improve the global efficiency of the system and therefore reduce fuel consumption and pollutant emissions.

Further research will focus on the identification of the best control strategy and shift speed for the lockup engagement and disengagement.

**Author Contributions:** D.D.: conceptualization, methodology, software, validation, supervision, investigation. G.R.: conceptualization, methodology, software, validation, supervision, investigation. F.A.: conceptualization, methodology, software, validation, supervision, investigation, writing—original draft preparation, writing—review and editing. All authors have read and agreed to the published version of the manuscript.

**Funding:** This research received no external funding.

**Institutional Review Board Statement:** Not applicable.

**Informed Consent Statement:** Not applicable.

**Data Availability Statement:** Not applicable.

**Conflicts of Interest:** The authors declare no conflict of interest.

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
