# Peer review of "Fuel Consumption Reduction and Efficiency Improvement in Urban Street Sweeper Using Power Split with Lockup Clutch Transmission"

_applsci, doi:10.3390/app121910160_

Round 1
Reviewer 1 Report
1. The contributions of this work should be added in the manuscript observably. In current version, the readers can not get the contributions of the paper easily.
2. For the results presented in the Figures 14-17, more explanations on them seem necessary and helpful to readers.
3.The literature review is not sufficient. Some recently papers about energy are suggested to be cited, such as:
[1] An optimal performance based new multi-objective model for heat and power hub in large scale users [J]. Energy, 2018, 161: 1234-1249.
Author Response
Manuscript ID: applsci-1943493
Title: Energy Assessment of a City Streetsweeper Through the Comparison of Different Transmission Layout: Efficiency and Fuel Consumption Evaluation
Dear Editor and Reviewers,
on behalf of all the authors, I would like to thank for your comments concerning the manuscript entitled “Energy Assessment of a City Streetsweeper Through the Comparison of Different Transmission Layout: Efficiency and Fuel Consumption Evaluation” (ID: applsci-1943493). Those comments are valuable and very helpful for revising and improving our paper.
The authors have tried their best to apply all comments in the revised article.
After the revision process, the authors think that the paper has been substantially improved, thanks to your valuable suggestions and observations.
Please check the attached revised version, which we would like to resubmit to Applied Sciences.
Best regards,
Fabio Alberti
Reviewer 1
- The contributions of this work should be added in the manuscript observably. In current version, the readers cannot get the contributions of the paper easily.
Answer: Thank you for your suggestion. The contribution of the work has been emphasized more in the introduction as can be read in lines 75-93.
- For the results presented in the Figures 14-17, more explanations on them seem necessary and helpful to readers.
Answer: Thank you for your comment. In order to have a greater overview of the transmissions, it have been tested also in a regulated cycle and the results have been inserted in results and discussion section. In addition, as suggested by the reviewer, more explanation have been added in this section. In particular all this part is highlighted in yellow, from line 321 to 374.
- The literature review is not sufficient. Some recently papers about energy are suggested to be cited, such as:
[1] An optimal performance based new multi-objective model for heat and power hub in large scale users [J]. Energy, 2018, 161: 1234-1249.
Answer: Thank you for your suggestions, the authors agree with the reviewer and have added the following references:
- [4] Ishaq, M.; Ishaq, H.; Nawaz, A. Life cycle assessment of electric scooters for mobility services: A green mobility solutions. Int. J. Energy Res. 2022, 1–18, doi:10.1002/er.8009.
- [5] Zhang, S.P.; Tak, T.O. Efficiency evaluation of electric bicycle power transmission systems. Sustain. 2021, 13, doi:10.3390/su131910988.
- [8] Ritari, A.; Vepsäläinen, J.; Kivekäs, K.; Tammi, K.; Laitinen, H. Energy consumption and lifecycle cost analysis of electric city buses with multispeed gearboxes. Energies 2020, 13, doi:10.3390/en13082117.
- [10] Kim, Y.S.; Kim, W.S.; Abu Ayub Siddique, M.; Baek, S.Y.; Baek, S.M.; Cheon, S.H.; Lee, S.D.; Lee, K.H.; Hong, D.H.; Park, S.U.; et al. Power transmission efficiency analysis of 42 kW power agricultural tractor according to tillage depth during moldboard plowing. Agronomy 2020, 10, doi:10.3390/agronomy10091263.
- [11] Siddique, M.A.A.; Baek, S.M.; Baek, S.Y.; Kim, W.S.; Kim, Y.S.; Kim, Y.J.; Lee, D.H.; Lee, K.H.; Hwang, J.Y. Simulation of fuel consumption based on engine load level of a 95 kw partial power-shift transmission tractor. Agric. 2021, 11, 1–17, doi:10.3390/agriculture11030276.
- [15] Ting, K.H.; Lee, L.S.; Pickl, S.; Seow, H.V. Shared mobility problems: A systematic review on types, variants, characteristics, and solution approaches. Appl. Sci. 2021, 11, doi:10.3390/app11177996.
- [16] Sun, Y.; Xu, J.; Lin, G.; Ni, F.; Simoes, R. An optimal performance based new multi-objective model for heat and power hub in large scale users. Energy 2018, 161, 1234–1249, doi:10.1016/j.energy.2018.07.173.
Reviewer 2 Report
The manuscript reported "Energy Assessment of a City Streetsweeper Through the Comparison of Different Transmission Layout: Efficiency and Fuel Consumption Evaluation". My detailed comments are listed below.
1- Tab. 1, Tab. 2, Eq. 1, Eq. 2 & Eq. 4: It is better not to use the parameter rpm to express the engine speed, which is easily confused with the unit rpm. In Tab. 1, 0,3556? “,”
2- 2.2 Powertrain layout: Why are the three layouts selected? Is there any other better layout?
3- Eq. 1: 3,6? It should be 3.6.
4- Eq. 4: ??????? It should be rpmICE.
5- Fig. 6, Fig.7 & Fig. 8: The quality of these figures is not so good.
6- Fig. 11: Please check the curves of 30%, 50% and 70%. The power is lower, so the force can’t reach 12000N.
7- Fig. 17: The two curves are very close. If we consider the complexity of the system and the overall mass, the conclusion is not convincing.
8- Lack of test verification, only simulation can’t explain the conclusion well.
Author Response
Manuscript ID: applsci-1943493
Title: Energy Assessment of a City Streetsweeper Through the Comparison of Different Transmission Layout: Efficiency and Fuel Consumption Evaluation
Dear Editor and Reviewers,
on behalf of all the authors, I would like to thank for your comments concerning the manuscript entitled “Energy Assessment of a City Streetsweeper Through the Comparison of Different Transmission Layout: Efficiency and Fuel Consumption Evaluation” (ID: applsci-1943493). Those comments are valuable and very helpful for revising and improving our paper.
The authors have tried their best to apply all comments in the revised article.
After the revision process, the authors think that the paper has been substantially improved, thanks to your valuable suggestions and observations.
Please check the attached revised version, which we would like to resubmit to Applied Sciences.
Best regards,
Fabio Alberti
Reviewer 2
- 1, Tab. 2, Eq. 1, Eq. 2 & Eq. 4: It is better not to use the parameter rpm to express the engine speed, which is easily confused with the unit rpm. In Tab. 1, 0,3556? “,”
Answer: Thank you, the authors have made the suggested changes
- 2 Powertrain layout: Why are the three layouts selected? Is there any other better layout?
Answer: Thank you for your comment. The adoption of the three layouts is presented is this work as an alternative of the full electric/hybrid technology. The main idea is that these types of transmissions combine the advantage of the mechanical gears in term of efficiency with the advantage of the hydraulic circuit in terms of power density. In fact, if we consider working vehicles, such as excavators, tractors or, like in this work, street sweepers, are usually equipped with conventional or hydrostatic transmission. The full electric/hybrid transmission could represent a better layout from an energetic point of view but, at the moment, this type of vehicles is far from the adoption of this technology, which is, on the other hand, in rapid growth in road cars. Also considering the small size and the slow speed of the street sweeper, the adoption of a full electric transmission can lead to a problem in terms of energy recover and so on.
- 1: 3,6? It should be 3.6.
Answer: Thank you, the authors have made the suggested changes
- 4: ??????? It should be rpmICE.
Answer: Thank you, rpmice has been appropriately corrected.
- 6, Fig.7 & Fig. 8: The quality of these figures is not so good.
Answer: Thank you for your comment. The quality of Figure 6, 7 and 8 has been improved.
- 11: Please check the curves of 30%, 50% and 70%. The power is lower, so the force can’t reach 12000N.
Answer: Thank you for your comment. As reported in line 289-292, the power partialization is not imposed to the system by a partilization of the engine power (which is constant and works at a constant speed of 1800 rpm) but applying a variable payload at the end of the transmission, after the final drive gear. This is made through a rotary friction which is totally closed for the maximum power (P_rated) and is partially open for the other cases. Basically, it works as a partial brake placed at the end of the transmission. Hence, at low speed, despite the rotary friction is partially open, it is not influent in terms of traction force. Increasing the speed on the contrary became influent and the traction force follows the trend shown in Figure 11.
- 17: The two curves are very close. If we consider the complexity of the system and the overall mass, the conclusion is not convincing.
Answer: Thank you for your comment. The authors agree with the reviewer’s comment. Related to this fact, in order to demonstrate the validity of the different architectures, it is also inserted a regulated cycle for testing both transmissions. As shown in Figure 17a and 17b in fact, the adoption of the lockup clutch leads to a consistent advantage in terms of efficiency and fuel consumption reduction during the working phase. It is necessary to observe that, if we consider the real cycle (mission test), only 1000 sec has been considered and of course, it seems that the fuel consumption reduction is less evident. However, if it is taken in consideration a daily working cycle projected, for example, in one year of work, and considering also an high number of streetsweepers operating in a urban center, the fuel consumption reduction and the pollutants emission reduction become considerable. This fact could justify the adoption of the lock up clutch.
- Lack of test verification, only simulation can’t explain the conclusion well.
Answer: Thank you for your comment. The design of this type of transmission represents a novelty concerning the application on a urban street sweeper. As also said before, usally this type of vehicles are equipped with conventional transmission or hydrostatic one. Hence, real data on the transmission are not available but is a future development if the results are reliable and interesting for some companies, for example, for producting a prototype. However, the transmission is tested through a real driving cycle.
Round 2
Reviewer 2 Report
The paper has been carefully revised. But There are several other problems.
1- Tab. 1: 0.3556 m? How about the accuracy? The accuracy should be millimeter.
2- Fig. 2: The maximum torque is 92.6 Nm at 1700 rpm. Why is it not shown in the figure?
3- Eq. 2: rpmmotor? Please check it.
Author Response
Response to the Reviewers’ Comments
Manuscript ID: applsci-1943493
Title: Energy Assessment of a City Streetsweeper Through the Comparison of Different Transmission Layout: Efficiency and Fuel Consumption Evaluation
Dear Editor and Reviewers,
on behalf of all the authors, I would like to thank for your comments concerning the manuscript entitled “Energy Assessment of a City Streetsweeper Through the Comparison of Different Transmission Layout: Efficiency and Fuel Consumption Evaluation” (ID: applsci-1943493). Those comments are valuable and very helpful for revising and improving our paper.
The authors have tried their best to apply all comments in the revised article.
After the revision process, the authors think that the paper has been substantially improved, thanks to your valuable suggestions and observations.
Please check the attached revised version, which we would like to resubmit to Applied Sciences.
Best regards,
Fabio Alberti
Reviewer 2
The authors would like to thanks the reviewer for the suggestions.
All the corrections have been inserted on the current version of the Manuscript and highlighted in yellow.